

# Background CO$_2$ levels and error analysis from ground-based solar absorption IR measurements in central Mexico

Jorge L. Baylon[1], Wolfgang Stremme[1], Michel Grutter[1], Frank Hase[2], and
Thomas Blumenstock[2]

[1]Centro de Ciencias de la Atmósfera, Universidad Nacional Autónoma de México, Mexico
[2]Institute of Meteorology and Climate Research, Karlsruhe Institute of Technology, Germany

*Correspondence to:* (baylon@atmosfera.unam.mx)

**Abstract.** In this investigation we analyze two common optical configurations to retrieve CO$_2$ total column amounts from solar absorption infrared spectra. The noise errors using either a KBr or a CaF$_2$ beamsplitter, a main component of a Fourier transform infrared (FTIR) spectrometer, are quantified in order to assess the relative precisions of the measurements. The configuration using a CaF$_2$

beamsplitter, as deployed by the instruments which contribute to the Total Carbon Column Observing Network (TCCON), shows a slightly better precision. However, we show that the precisions in X$_{CO_2}$ retrieved from >96% of the spectra measured with a KBr beamsplitter, fall well below 0.2%. A small bias in X$_{CO_2}$ (KBr - CaF$_2$) of +0.56 ± 0.25 ppm was found when using an independent data set as reference. This value, which corresponds to +0.14 ± 0.064 %, is slightly larger than the mean

precisions obtained and could be taken into account when homogenizing or comparing data from both beamsplitters. A 3-year X$_{CO_2}$ time series from FTIR measurements at the high-altitude site of Altzomoni in central Mexico presents clear annual and diurnal cycles and a trend of +2.2 ppm/yr could be determined.

## 1 Introduction

During the last decades, carbon dioxide (CO$_2$) has exceeded the pre-industrial levels by about 40% mainly due to fossil fuel combustion and land use change (Hartmann et al., 2013), contributing more than any other gas to the positive total radiative forcing of the Earth and becoming the most important anthropogenic greenhouse gas (Myhre et al., 2013). The quantification of the spatial distribution and temporal variation of CO$_2$ sources and sinks can help to understand the anthropogenic

contributions of CO$_2$ to the carbon cycle. This task can be achieved by monitoring the atmosphere using ground-based and satellite observations. Ground-based networks like the Global Atmospheric Watch WMO (2014) or the Total Column Carbon Observing Network (TCCON) provide data sets of CO$_2$ concentrations around the world. TCCON is a network of Fourier transform infrared spec-



trometers that record solar absorption spectra in the near infrared (NIR) spectral region in order to
retrieve column-averaged dry-air mole fractions of $CO_2$ ($X_{CO_2}$) and other molecules that absorb
in the NIR (Wunch et al., 2011). TCCON aims to provide reliable, long-term validation data sets
for satellite measurements and to improve current knowledge of the carbon cycle. Measurements of
$CO_2$ from space have been done by many satellite missions like ACE (Foucher et al., 2011), AIRS
(Chahine et al., 2008), IASI (Crevoisier et al., 2009), TES (Kulawik et al., 2010), SCIAMACHY
(Reuter et al., 2011), GOSAT (Kuze et al., 2009) and OCO-2 (Wunch et al., 2016) with the last three
missions relying on TCCON data for validation.

Studies done to estimate the $CO_2$ concentrations in Mexico City and central Mexico have been
scarce, the first one was conducted during 1981 and 1982 in which the diurnal and seasonal variation
was estimated taking air samples in different parts of the city (Báez et al., 1988). In two different
campaigns (MCMA-2003 and MILAGRO) the $CO_2$ fluxes and concentrations during typical days in
the Mexico City Metropolitan Area (MCMA) were estimated using the eddy covariance technique
(Velasco et al., 2005, 2009). The first study using a FTS was done during September 2001 in the
south of the MCMA using a Nicolet Nexus interferometer with a resolution of 0.125 $cm^{-1}$ and
retrieving $CO_2$ in the 723-766 $cm^{-1}$ spectral region (Grutter, 2003). From this study the variability
and average diurnal cycle of $CO_2$ was recorded with high temporal resolution.

The Altzomoni site is located to the southeast of the MCMA at a height of almost 4,000 m above
the sea level. Measurements of NIR and MIR spectra have been conducted since 2012 using a Bruker
IFS 120/5 HR. The site is part of the NDACC network since 2015 and has been reporting vertical
columns of $O_3$, CO, $N_2O$ and $CH_4$ among other gases. As part of the NDACC instrumental config-
uration, a KBr beamsplitter has been used for most of the measurements but for a limited number of
days, a $CaF_2$ beamsplitter was also used in order to meet the TCCON instrumental requirements and
compare the effect of each configuration in the retrievals of $CO_2$, $O_2$ as well as in the estimation of
$X_{CO_2}$.

This paper presents $X_{CO_2}$ retrieved from NIR spectra measured from December 2012 to Decem-
ber 2015 in the Altzomoni site and an intercomparision of how the use of KBr and $CaF_2$ beamsplit-
ters affect the errors and precision of $CO_2$, $O_2$ columns and $X_{CO_2}$ mole fractions. Section 2 describes
the configuration used in the measurement of the NIR spectra and how these spectra were analyzed
for producing the time series of $X_{CO_2}$. An evaluation of how the used beamsplitter influence the total
column retrievals and the calculation of $X_{CO_2}$ by means of an estimation of noise errors, precision
and bias of each configuration is presented in Sect. 3. The characterisitc seasonal and diurnal cycles,
as well as the observed trend for this sub-tropical site in central Mexico, are presented in Sect. 4.



## 2 FTIR instrument, measurement site and spectral analysis

A high resolution Fourier Transform InfraRed (FTIR) spectrometer, Bruker model HR 120/5, was
deployed to measure solar absorption spectra under clear sky conditions. The instrument begun op-
erations at a high-altitude location in central Mexico in 2012 as part of a collaboration between the
National Autonomous University of Mexico (UNAM) and the Karlsruhe Institute of Technology
(KIT) and since 2015, Altzomoni is part of the Network for the Detection of Atmospheric Composi-
tion Change (NDACC). Routine and remotely-operated measurements are performed with a spectral
resolution of 0.005 cm$^{-1}$ using a KBr beamsplitter, a set of band-pass filters and liquid-nitrogen
cooled MCT and InSb detectors, according to NDACC specifications. The solar tracking is based on
the Camtracker system (Gisi et al., 2011) used in other NDACC and TCCON sites and is housed in
a dome which can be operated remotely.

Alternatively, an InGaAs detector is used to record near-infrared (NIR) spectra within each mea-
surement sequence with a resolution of 0.02 cm$^{-1}$. These NIR spectra, used in this study to retrieve
$CO_2$ and $O_2$, were recorded as the average of 2 scans taking approximately 38 seconds with a scan-
ner speed of 40 kHz. In each measurement sequence, a set of six NIR measurements are recorded,
taking around 5 minutes. This small change in solar angles allow the consideration that all measure-
ments belong to the same airmass. A $CaF_2$ beamsplitter is also available and was used for a small
number of days for the purpose of estimating the noise levels in each optical configuration and how
this affects the retrievals.

The FTIR instrument is located at the Altzomoni high-altitude station (19.1187°N, 98.6552°W)
located in central Mexico within the Izta-Popo National Park, 60 km southeast of Mexico City, at an
altitude of 3,985 m a.s.l. This station is part of the University Network of Atmospheric Observatories
(www.ruoa.unam.mx) and comprises a complete set of in situ and meteorological instrumentation.

The measured spectra were analyzed with the retrieval code PROFFIT which uses the radiative
transfer code PROFFWD (Hase et al., 2004). For the calculation of the dry-air column-average mole
fractions of carbon dioxide $X_{CO_2}$, $CO_2$ and $O_2$ were retrieved separately using a profile scaling
procedure and with the microwindows and interfering species listed in Table 1. Pressure and tem-
perature profiles from the National Center for Environment Prediction (NCEP) were used and the a
priori profiles were obtained from the Whole Atmosphere Community Climate Model (WACCM).
A single a priori profile of each retrieved species was used for the entire set of measurements.

## 3 Effect of the beamsplitter on the retrieval

The TCCON instrumental requirements state that the network's necessary precision is best achieved
using a $CaF_2$ beamsplitter (TCCON-Wiki), but in the case of the Altzomoni site, this would mean
sacrificing routine measurements of spectra in the mid-infrared (MIR) region since the beamsplitter
change needs to be performed manually. Given the location of the site and aside from complying with





a long term commitment with NDACC, it is of great interest to perform measurements of gases which absorb in the MIR region in order to characterize pollution transport events in the region and study the composition of the gases emitted by the active Popocatépetl volcano. For these reasons, a KBr

beamsplitter has been part of the configuration used in the site and the use of the $CaF_2$ beamsplitter has been limited. Figure 1 shows two tungsten NIR lamp spectra, one measured with KBr (red) and another one with $CaF_2$ (blue) to illustrate the features of each measurement in the NIR region. The KBr spectra shows a slightly lower intensity and a dip on the 5000-6000 $cm^{-1}$ spectral region. Kiel et al. (2016) showed that a small curvature might affect the retrieval results if no baseline is adjusted.

The settings used for the retrievals of this study adjusted a smoothed baseline with 20 parameters for each microwindow used. It's worth mentioning that the effects from the baseline curvature in the spectral region around the dip introduced by the KBr beamsplitter are removed using a simplified radiometric calibration, assuming that the tungsten lamp produces a blackbody spectrum (T=1700 K) and that there is no self-emission of the optical set-up in the spectral region above 4000 $cm^{-1}$.

In order to compare the impact of the KBr and $CaF_2$ beamsplitters on the retrievals and since far more measurements are available with the KBr beamsplitter (27,148) than with $CaF_2$ (2,093), an ensemble of KBr measurements was formed reproducing the size and solar zenith angle (SZA) distribution of the $CaF_2$ measurements. A condition imposed was to only consider sets of consecutive measurements done within a five minute lapse so that the precision of each retrieval product and SZA

could be calculated. As shown in Fig. 2, the KBr ensemble consisted in measurements from 101 days between July 30, 2013 and December 30, 2015 while the $CaF_2$ ensemble had measurement from 43 days from February 15, 2014 to June 23, 2015.

For the case of $CO_2$ column measurements, two references of precision exist: Rayner and O'Brien (2001) showed that a 0.25% network precision would improve the current knowledge of the carbon

cycle while Olsen and Randerson (2004) suggested that a 0.1% precision would allow to assess the strength of the carbon sink in the Northern Hemisphere. The following sub-sections are dedicated to determine where does the Altzomoni data fall, using routinely a KBr beamsplitter, in terms of these two benchmarks.

### 3.1 Retrieved $CO_2$ and $O_2$ error budgets

The error calculation implemented in PROFFIT allows to estimate the errors associated with a total column retrieval (Barthlott et al., 2015). These include channelling and offset, instrumental line shape (ILS), temperature profile, line-of-sight (LOS), solar lines, spectroscopy and noise errors. The errors were calculated for each of the measurements that comprise the KBr and $CaF_2$ ensembles. The magnitude of the uncertainties and the statistical and systematic contributions for each source are

listed in Table 2. The purpose of obtaining the errors from the PROFFIT software was to determine the value of noise present in each measurement and how it depends on the beamsplitter used. The noise calculation from PROFFIT takes into account the derivatives of the retrieval with respect of the





measurement and the difference between the measurement and a simulated spectra from the forward model. Figures 3 and 4 show how the mean noise error from the $CO_2$ and the $O_2$ total columns

depend on the beamsplitter and the SZA, while Tables 3 and 4 show the mean values of each error source for two SZA values (20 and $70°$). The noise error of the $CO_2$ column shows good agreement between beamsplitters for SZA's above $30°$, but is lower in KBr measurements at smaller angles. For the $O_2$ column, the errors have similar behavior for angles below $30°$, but the noise errors for angles above this value remain more or less constant and are 30 - 40 % larger for KBr than for

$CaF_2$. Overall, the total statistical and systematic errors for both beamsplitters estimated with this technique are quite similar.

### 3.2 Statistical precision from consecutive measurements

For a statistical estimation of the precision, we consider that the standard deviation of the consecutive measurements done within a 5-minute lapse (typically 6 spectra) represents the overall precision of

the measurements. This method is based on the assumption that the actual gas columns undergo smaller changes in the short time considered than the measurement error. With the three products derived from a NIR measurement ($CO_2$ and $O_2$ columns and $X_{CO_2}$), three different precisions were calculated and used for estimating which part of the random error is independent from the $CO_2$ and $O_2$ columns and which part is correlated and thus cancels out when the $X_{CO_2}$ ratio is calculated.

Assuming that the precisions of $CO_2$ and $O_2$ ($\sigma_{CO_2}$ and $\sigma_{O_2}$) are due to both the noise and the correlated errors (see Eq. 1&2), and the precision of $X_{CO_2}$ depends only in the noise from both columns (Eq. 3), a system of equations was formed and solved using the mean precisions of the three products to obtain the mean correlated and noise errors of $CO_2$ and $O_2$ for each beamsplitter.

$$(\sigma_{CO_2})^2 = (\sigma^{Correlated})^2 + (\sigma_{CO_2}^{Noise})^2 \tag{1}$$

$$(\sigma_{O_2})^2 = (\sigma^{Correlated})^2 + (\sigma_{O_2}^{Noise})^2 \tag{2}$$

$$(\sigma_{X_{CO_2}})^2 = (\sigma_{CO_2}^{Noise})^2 + (\sigma_{O_2}^{Noise})^2 \tag{3}$$

As can be seen in the results from this exercise presented in Table 5, the mean noise errors from the columns are within the range of the values obtained using PROFFIT and in the case of $X_{CO_2}$, the mean value from all the KBr measurements in the ensamble was 25% higher than with $CaF_2$.

The contribution appears to be dominated by the noise in the $O_2$ retrieval. This is in accordance to the result in Sect. 3.1. However as Figure 5 shows, the mean $X_{CO_2}$ precisions obtained from both beamsplitters were below 0.1% and those of >96% of all the spectra in the ensemble measured with a KBr beamsplitter, fall below the 0.2% value.

### 3.3 Bias estimation

The systematic difference between beamsplitters was estimated for the three products obtained. Since there are no temporal coincidences between the measurements with both beamsplitters, an



independent set of measurements was used to calculate the bias. The ensembles were sorted in bins, so that the mean values of the data in each bin could be compared even if the measurements on the ensembles weren't coincident in time.

For $X_{CO_2}$, the continuous data set of $CO_2$ in situ measurements from the Mauna Loa Observatory (MLO; 19.5362°N, 155.5763°W, 3,397 m.a.s.l) was chosen, given the fact that both sites share a similar latitude and altitude. An in situ measurement in Altzomoni is now available but the data does not cover the entire period of the FTIR ensamble. The MLO (in situ) and Altzomoni (FTIR) data sets are in good agreement, as shown in Fig 6, and both present similar seasonal cycles dominating
the observed variability. However, the amplitude in the MLO time series is approximately 5 ppm larger, possibly due to the different location and that Altzomoni is measuring total columns and not surface concentrations. There were 189 coincidences for the KBr and 174 for $CaF_2$ data sets. The bias was obtained from the mean of the KBr−$CaF_2$ differences of these coincidences, sorted in 13 bins generated using the measurements in MLO. From the correlation plot for each beamsplitter and
a Bland-Altman plot (Bland and Altman, 1986) for their differences shown in Figure 7, a bias of $+0.14 \pm 0.064$ % is obtained for $X_{CO_2}$.

In the case of $O_2$ total columns, a bias was calculated by estimating the dry pressure column from surface pressure measurements at Altzomoni and the $H_2O$ total columns retrieved in the NIR spectral region, which in turn was multiplied by the factor 0.2095 to convert to $O_2$ column. The number of
coincidences obtained between the data sets was 100 for KBr and 110 for $CaF_2$. Figure 8 shows the plots and KBr−$CaF_2$ differences resulting in a bias of $-0.17 \pm 0.029$ %.

The bias for $CO_2$ column was calculated from the biases obtained above ($\Delta X_{CO_2}$ and $O_2$ $\Delta O_2$) using Eq. 4:

$$\Delta X_{CO_2} = \frac{\partial X_{CO_2}}{\partial CO_2} \cdot \Delta CO_2 + \frac{\partial X_{CO_2}}{\partial O_2} \cdot \Delta O_2, \qquad (4)$$

from which a value of $\Delta CO_2 = -0.030 \pm 0.070$ % is obtained. The biases and the mean values for each beamsplitter are summarized in Table 6. For the homogenization of a data set, the relative $X_{CO_2}$ bias between beamsplitters can be corrected by multiplying the KBr data by the factor 0.9986.

## 4    Observed time series

Figure 9 shows the daily means of the $X_{CO_2}$ in black, derived from 29,241 measurements done in
Altzomoni during 510 days between December 28, 2012 and December 30, 2015. A function was adjusted to the data using Eq. 5, taken from Wunch et al. (2013), where $x$ is the decimal year and the obtained fitting parameters were as follows: $\alpha = 2.19$ ppm yr$^{-1}$, $a_0 = -0.0040$ ppm, $a_1 = -0.93$ ppm, $a_2 = 0.95$ ppm, $b_1 = 1.60$ ppm, $b_2 = -0.54$ ppm. The linear term determines the trend of the series which has the value of 2.2 ppm/year. The same function fitted over the MLO data set shows





also a 2.2 ppm/year trend.

$$f(x) = \alpha x + \sum_{k=0}^{2} a_k cos(2\pi kx) + b_k sin(2\pi kx). \tag{5}$$

The daily average mixing ratio from all the FTIR data is presented in Figure 10. These hourly data were detrended using the fitted curve obtained for the annual cycle and the local time was converted to true solar time in order to observe the distinct effect of vegetation. The concentration of $CO_2$ is

lowered on average during the day up to 1.5 ppm through photosynthesis. The weekday averages were also calculated and no distinct weekly pattern was detected from these data which indicates that the measurements are representative for the free atmosphere and the influence of the nearby cities are minimal with respect to the total column. For comparison, the detrended diurnal cycle of the MLO data is also shown in the figure, showing a similar behavior but with a smaller amplitude

and a minimum which occurs 3 hours later.

## 5 Conclusions

Solar absorption FTIR measurements done with KBr and $CaF_2$ beamsplitters were compared using equivalent ensembles containing more than 2 thousand spectra. The two methods used for evaluating the statistical errors gave similar results. In the case of $CO_2$ columns, the noise levels from the KBr

measurements are on average 20% lower than from $CaF_2$ measurements when solar zenith angles are below 30°. Measurements with larger SZA's have similar errors with both beamsplitters. Larger error differences are encountered from the $O_2$ column retrievals. For angles below 30°the noise in KBr measurements is around 29% lower but increases with the angle and remains constant above the $CaF_2$ levels, approximately 38% higher.

Thus, in this study an estimation of the precision of each ensamble shows that the largest statistical error contribution in $X_{CO_2}$ comes from the $O_2$ column retrieval. This outcome has the implication that column averaged mixing ratios retrieved using KBr beamsplktters have noise-related errors which are on average about 25% larger than with $CaF_2$. However, mean $X_{CO_2}$ precisions was found to be below 0.1% and >96% of the measurements made with both optical configurations fall well

below the 0.2% precision.

These results provide enough evidence that measurements performed with a KBr beamsplitters are reliable and useful for carbon cycle studies. This includes all FTIR instruments which are committed to comply with NDACC requirements and have an additional InGaAs detector available for NIR spectral measurements. A larger number of sites producing confident $X_{CO_2}$ data sets would allow to

increase our current knowledge of the variability of this important greenhouse gas.

When doing direct comparisons across a network or using single retrievals for intercomparing timed observations, however, one needs to be cautious and consider a possible bias. We have estimated a bias of 0.14% for $X_{CO_2}$ between beamsplitters using data from the Mauna Loa Observatory.





A rich data set of $X_{CO_2}$ was put together from more than 3 years of measurements in central
Mexico. A very distinct annual cycle was identified with an amplitude of ~6 ppm and a positive
trend of 2.2 ppm/year, while the mean diurnal pattern reflects a decrease of up to ~1.5 ppm during
sunlit hours. This data set confirms a small influence of anthropogenic emissions especially during
the afternoon, when the regional boundary layer reaches the height of the station.

*Acknowledgements.* We would like to thank the financial support from UNAM-DGAPA (IN109914 & IN112216)
and CONACYT (0249374 & 0239618). JB received a full stipend from CONACYT throughout his PhD work
as well as financial support from UNAM's Earth Sciences Graduate Program. The Mauna Loa Observatory and
the RUOA Network (Red Universitaria de Observatorios Atmosféricos – UNAM) are acknowledged for making
the in situ measurements available. Special thanks go to Alejandro Bezanilla, Maria Eugenia González, Delibes
Flores, Héctor Soto and Omar López for their technical support in this investigation.



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





**Table 1.** Microwindows and interfering species used for $CO_2$ and $O_2$ retrievals.

|  | Microwindows ($cm^{-1}$) | Interfering species |
|---|---|---|
| $CO_2$ | 6180.0 – 6260.0, 6310.0 – 6380.0 | $H_2O$, $CH_4$ |
| $O_2$ | 7765.0 – 8005.0 | $H_2O$, $CO_2$ |

in urban Mexico City, Atmospheric Chemistry and Physics, 9, 7325–7342, doi:10.5194/acp-9-7325-2009, http://www.atmos-chem-phys.net/9/7325/2009/, 2009.

WMO: The Global Atmospheric Watch Programme: 17th WMO/IAEA Meeting on Carbon Dioxide, Other Greenhouse Gases and Related Tracers Measurement Techniques (GGMT-2013), Beijing, China, 10-13 June 2013, GAW Report No. 213, World Meteorological Organization, Geneva, Switzerland, 2014, 2014.

Wunch, D., Toon, G. C., Blavier, J.-F. L., Washenfelder, R. A., Notholt, J., Connor, B. J., Griffith, D. W. T., Sherlock, V., and Wennberg, P. O.: The Total Carbon Column Observing Network, Philosophical Transactions of the Royal Society of London A: Mathematical, Physical and Engineering Sciences, 369, 2087–2112, doi:10.1098/rsta.2010.0240, http://rsta.royalsocietypublishing.org/content/369/1943/2087, 2011.

Wunch, D., Wennberg, P. O., Messerschmidt, J., Parazoo, N. C., Toon, G. C., Deutscher, N. M., Keppel-Aleks, G., Roehl, C. M., Randerson, J. T., Warneke, T., and Notholt, J.: The covariation of Northern Hemisphere summertime $CO_2$ with surface temperature in boreal regions, Atmospheric Chemistry and Physics, 13, 9447–9459, doi:10.5194/acp-13-9447-2013, http://www.atmos-chem-phys.net/13/9447/2013/, 2013.

Wunch, D., Wennberg, P. O., Osterman, G., Fisher, B., Naylor, B., Roehl, C. M., O'Dell, C., Mandrake, L., Viatte, C., Griffith, D. W., Deutscher, N. M., Velazco, V. A., Notholt, J., Warneke, T., Petri, C., De Maziere, M., Sha, M. K., Sussmann, R., Rettinger, M., Pollard, D., Robinson, J., Morino, I., Uchino, O., Hase, F., Blumenstock, T., Kiel, M., Feist, D. G., Arnold, S. G., Strong, K., Mendonca, J., Kivi, R., Heikkinen, P., Iraci, L., Podolske, J., Hillyard, P. W., Kawakami, S., Dubey, M. K., Parker, H. A., Sepulveda, E., Rodriguez, O. E. G., Te, Y., Jeseck, P., Gunson, M. R., Crisp, D., and Eldering, A.: Comparisons of the Orbiting Carbon Observatory-2 (OCO-2) $X_{CO_2}$ measurements with TCCON, Atmospheric Measurement Techniques Discussions, 2016, 1–45, doi:10.5194/amt-2016-227, http://www.atmos-meas-tech-discuss.net/amt-2016-227/, 2016.



**Table 2.** Error sources used in the error estimation, the second column gives the uncertainty used and the third the statistical (Stat) and systematic (Sys) contributions of each source in percentage.

| Error Source | Uncertainty | Stat / Sys [%] |
|---|---|---|
| Baseline (offset / channelling) | 0.1% / 0.2% | 50 / 50 |
| ILS (mod. eff. / phase error) | 2% / 0.01 rad | 50 / 50 |
| Line of sight | 0.001 rad | 90 / 10 |
| Solar lines (intensity / scale) | 1% / 1e10$^{-6}$ | 80 / 20 |
| Temperature | 1, 2 & 5 K | 70 / 30 |
| Spectroscopic parameters ($S$ / $\gamma$) | 2% / 5% | 0 / 100 |
| Measurement noise | - | 100 / 0 |

**Table 3.** Mean ensemble values of the statistical (Stat) and systematic (Sys) errors for $20°$ and $70°$ SZA (given in %), in $CO_2$ columns due to the assumed error sources of Table 2.

| $CO_2$ | SZA = $20°$ | | | | SZA = $70°$ | | | |
|---|---|---|---|---|---|---|---|---|
| | KBr | | $CaF_2$ | | KBr | | $CaF_2$ | |
| | Stat | Sys | Stat | Sys | Stat | Sys | Stat | Sys |
| Baseline | 0.082 | 0.082 | 0.081 | 0.081 | 0.085 | 0.085 | 0.085 | 0.085 |
| ILS | 0.073 | 0.073 | 0.076 | 0.076 | 0.043 | 0.043 | 0.043 | 0.043 |
| LOS | 0.031 | 0.0035 | 0.031 | 0.0034 | 0.24 | 0.026 | 0.24 | 0.026 |
| Solar lines | 0.0071 | 0.0018 | 0.0092 | 0.0023 | 0.0065 | 0.0016 | 0.0082 | 0.0020 |
| Temperature | 0.019 | 0.0081 | 0.012 | 0.0053 | 0.031 | 0.013 | 0.032 | 0.014 |
| Spectroscopy | — | 2.13 | — | 2.16 | — | 2.13 | — | 2.13 |
| Noise | 0.039 | — | 0.047 | — | 0.037 | — | 0.035 | — |
| TOTAL | 0.12 | 2.14 | 0.13 | 2.16 | 0.26 | 2.13 | 0.26 | 2.13 |





**Table 4.** Mean ensemble values of the statistical (Stat) and systematic (Sys) errors for 20° and 70° (given in %), in $O_2$ columns due to the assumed error sources of Table 2.

| $O_2$ | SZA = 20° | | | | SZA = 70° | | | |
|---|---|---|---|---|---|---|---|---|
| | KBr | | CaF$_2$ | | KBr | | CaF$_2$ | |
| | Stat | Sys | Stat | Sys | Stat | Sys | Stat | Sys |
| Baseline | 0.090 | 0.090 | 0.087 | 0.087 | 0.089 | 0.089 | 0.090 | 0.090 |
| ILS | 0.060 | 0.060 | 0.059 | 0.059 | 0.032 | 0.032 | 0.032 | 0.032 |
| LOS | 0.031 | 0.0034 | 0.030 | 0.0033 | 0.22 | 0.025 | 0.22 | 0.024 |
| Solar lines | 0.0077 | 0.0019 | 0.0077 | 0.0019 | 0.0044 | 0.0011 | 0.0047 | 0.0012 |
| Temperature | 0.029 | 0.012 | 0.033 | 0.014 | 0.031 | 0.013 | 0.032 | 0.014 |
| Spectroscopy | — | 2.12 | — | 2.10 | — | 2.89 | — | 2.90 |
| Noise | 0.046 | — | 0.063 | — | 0.074 | — | 0.053 | — |
| TOTAL | 0.13 | 2.12 | 0.13 | 2.10 | 0.25 | 2.89 | 0.25 | 2.90 |

**Table 5.** Mean precision, noise and correlation errors (given in %) for both ensembles, of 2,093 measurements each, using KBr and CaF$_2$ beamsplitters. The $X_{CO_2}$ precision is the root-sum-square of the noise errors of $CO_2$ and $O_2$.

| | $\sigma_{CO_2}$ | $\sigma_{O_2}$ | $\sigma^{Correlated}$ | $\sigma_{CO_2}^{Noise}$ | $\sigma_{O_2}^{Noise}$ | $\sigma_{XCO_2}$ |
|---|---|---|---|---|---|---|
| KBr | $0.072 \pm 0.0041$ | $0.098 \pm 0.0042$ | $0.055 \pm 0.0055$ | $0.047 \pm 0.0065$ | $0.082 \pm 0.0037$ | $0.094 \pm 0.0033$ |
| CaF$_2$ | $0.078 \pm 0.0046$ | $0.090 \pm 0.0037$ | $0.065 \pm 0.0031$ | $0.043 \pm 0.0048$ | $0.062 \pm 0.0033$ | $0.075 \pm 0.0022$ |

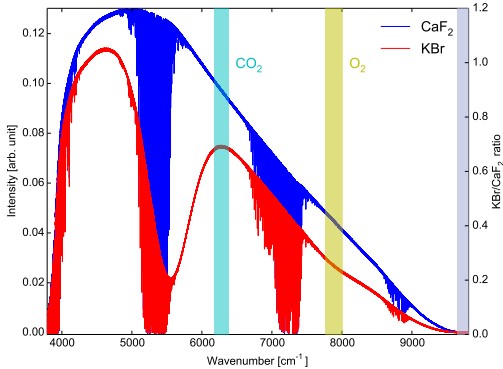

**Figure 1.** Spectra of a near infrared (NIR) lamp measured with KBr (red) and CaF$_2$ (blue) beamsplitters, the spectral regions where the $CO_2$ and $O_2$ target gases are retrieved are shown in cyan and yellow, respectively. The KBr/CaF$_2$ lamp intensity ratio, as shown in grey, is smooth at the target regions and is lower for $O_2$ (the gas which presents larger error differences among beamsplitters).



**Table 6.** Mean values of $X_{CO_2}$, $CO_2$ and $O_2$ and the bias obtained for each of them using the beamsplitters ensembles. For $X_{CO_2}$, the Mauna Loa Observatory (MLO) was used and for $O_2$ the dry pressure column was calculated and multiplied by 0.2095. The bias of $CO_2$ was obtained from Eq. 4.

| | KBr | CaF$_2$ |
|---|---|---|
| **$X_{CO_2}$** | | |
| Coincidences | 189 | 174 |
| Mean Value FTIR [ppm] | 396.07 ± 0.12 | 396.68 ± 0.14 |
| Mean Value MLO [ppm] | 399.82 ± 0.14 | 402.04 ± 0.20 |
| Mean Difference [ppm] | -3.75 ± 0.077 | -5.36 ± 0.11 |
| Bias (KBr-CaF$_2$) [ppm] | +0.56 ± 0.25 (+0.14 ± 0.064 %) | |
| **$O_2$** | | |
| Coincidences | 100 | 110 |
| Mean Value FTIR [$10^{24}$ molec cm$^{-2}$] | 2.90 ± 0.00089 | 2.90 ± 0.00077 |
| Mean Value dry pressure column [$10^{24}$ molec cm$^{-2}$] | 2.82 ± 0.00055 | 2.82 ± 0.00046 |
| Mean Difference [$10^{24}$ molec cm$^{-2}$] | 0.078 ± 0.00050 | 0.081 ± 0.00051 |
| Bias (KBr-CaF$_2$) [$10^{24}$ molec cm$^{-2}$] | -0.0050 ± 0.00083 (-0.17 ± 0.029 %) | |
| **$CO_2$** | | |
| Mean Value FTIR [$10^{24}$ molec cm$^{-2}$] | 5.49 ± 0.00034 | 5.50 ± 0.00031 |
| Bias [$10^{24}$ molec cm$^{-2}$] | -0.0016 ± 0.0039 (-0.030 ± 0.070 %) | |

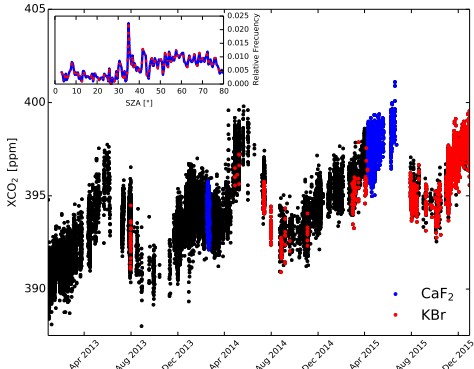

**Figure 2.** Time series of the $X_{CO_2}$ data set from Altzomoni site (black points) with the elements of the KBr (red points) and the CaF$_2$ (blue points) ensembles. The inset plot show the SZA distribution of both ensembles.





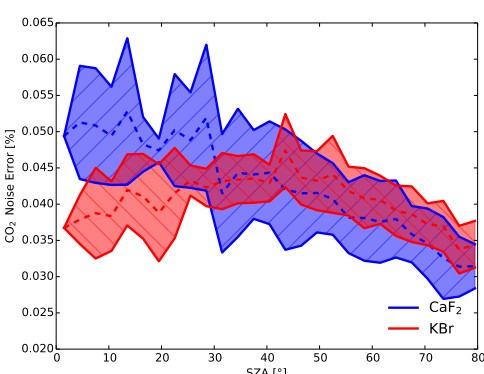

**Figure 3.** Mean noise error from PROFFIT for $CO_2$ total column in function of SZA.

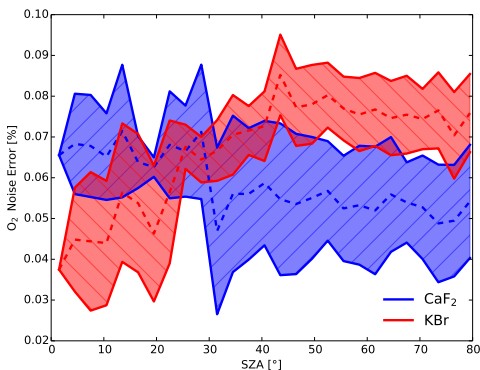

**Figure 4.** Mean noise error from PROFFIT for $O_2$ total column in function of SZA.





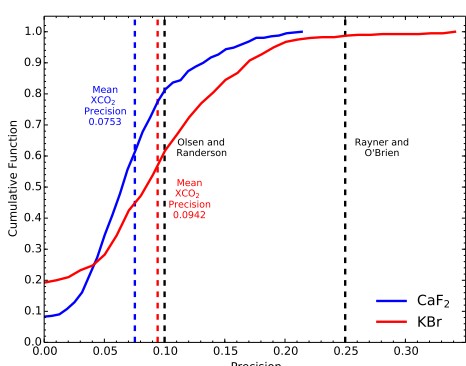

**Figure 5.** Cumulative function of $X_{CO_2}$ precision for both beamsplitters with the blue and red dashed lines denoting the precision values from Table 5 obtained from the top-down approach. The black line depicts the precision goal from TCCON.

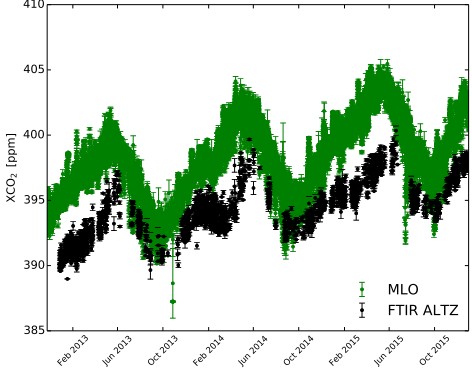

**Figure 6.** Hourly means of Altzomoni $X_{CO_2}$ (FTIR ALTZ, black points) and Mauna Loa Observatory in situ $CO_2$ (MLO, green points)





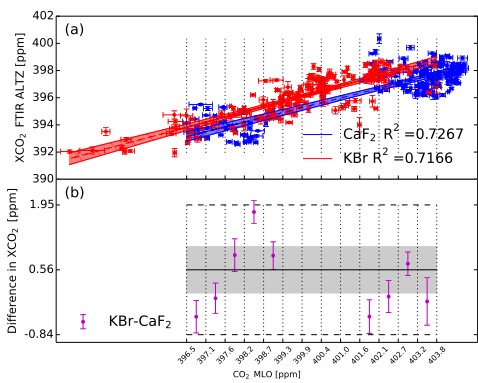

**Figure 7.** Upper panel (a) shows the coincidences between the hourly means of the $X_{CO_2}$ from the KBr (red) and CaF$_2$ (blue) ensembles and the $CO_2$ from the Mauna Loa Observatory (MLO) data set with a linear regression of the two sets with the shaded area representing a 95% confidence interval. Lower panel (b) shows the difference of means of KBr and CaF$_2$ (purple points) for each bin (vertical lines) in a Bland-Altman plot, with the black dashed lines showing the standard deviation of all points. The black solid line represents the bias and the shaded area the standard error of the bias, both reported in Table 6.

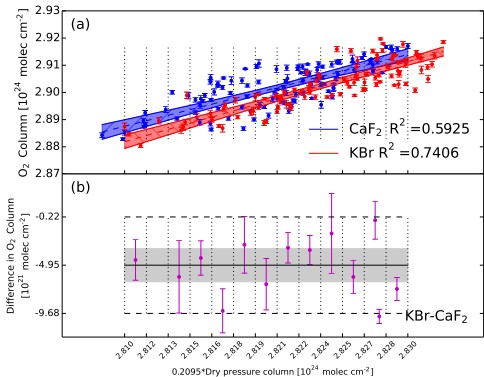

**Figure 8.** Upper panel (a) shows the coincidences between the hourly means of the O$_2$ from the KBr (red) and CaF$_2$ (blue) ensembles and the O$_2$ column obtained from the dry pressure column with a linear regression of the two sets with the shaded area representing a 95% confidence interval. Lower panel (b) shows the difference of means of KBr and CaF$_2$ (purple points) for each bin (vertical lines) in a Bland-Altman plot, with the black dashed lines showing the standard deviation of all points. The black solid line represents the bias and the shaded area the standard error of the bias, both reported in Table 6.



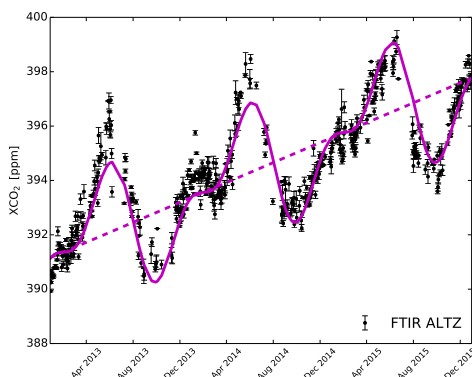

**Figure 9.** Daily means of the $X_{CO_2}$ data set from the Altzomoni site (black points). The purple solid line is a curve adjusted to the series (see text) with a linear term, represented by the dashed line, of 2.2 ppm/year which is the trend for this 3-year period.

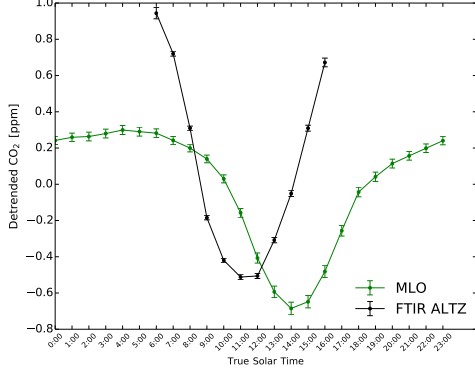

**Figure 10.** Detrended diurnal cycle of the $X_{CO_2}$ data set from Altzomoni site (black line) with detrended diurnal cycle of MLO (green line). The time of each measurement was converted to true solar time. Each bin represents the average of the hour marked containing data from 0 to 59 minutes.