# Peer review of "Background CO2 levels and error analysis from ground-based solar absorption IR measurements in central Mexico"

_Atmospheric Measurement Techniques, 2016_

## Referee Comment (RC1) · Anonymous Referee #1 · 2 Feb 2017

This paper reports on an important observation in the area of global climate change and of the techniques used to determine change. The paper is well written. There is a clear objective to be addressed, it is well defined and sufficiently limited in scope to be addressed in a short manuscript. The methods used by the authors to treat errors in the observations are solid and well described. The paper is clear and concise. I present two questions regarding the technique and some small points that might improve the presentation: 1) Is there some reference that supports the validity of comparing the total column amount of CO2, XCO2, with the in situ CO2 amount , as shown in Figure 6. I believe that when averaging kernels and error covariances are taken into account there is little valid overlap. See Rodgers and Connor, JGR, 108, 4116, 2003. 2) Since

one of the main objectives of the work is to compare the error contributions of KBr and CaF2 beamsplitters, which of the error sources in Table 2 might have some physical justification for being different between the two beamsplitters. In Table 3 the largest differences in error for CO2 between the splitters seems to be in the noise and solar line contributions. Is there a reason for this? Minor points

Line 7 define XCO2 Line 17 water vapor contributes more than any other gas to radiative forcing Line 24 it might be good to add some numbers to describe where the near infrared is, various readers may have various ideas about what is near or far (also line 90) Line55 /characterisitc/characteristic/ Line 59 /begun/began/ Line 62 /is part/has been part/ Line 104 comment on how effective the removal of the baseline curvature is wrt the retrieval Line 115 /to assess/an assessment of/ Line 117 /determine where does/determining where/ Line 120 /allows to/allows one to/ Line 154 /ensamble/ensemble/ also in Line 168,215 Line 246 2 also Line 257, 287 Line 254 n/a-n/a also Line 301, 310

Figure 5 which black line is the TCCON goal .1 or .25

---

## Referee Comment (RC2) · P. O. Wennberg (Referee) · 4 Feb 2017

Baylon et al. present a nice set of measurements of XCO2 from an FTS located at the high-altitude UNAO station near Mexico City. Measurements obtained with two different FTS beamsplitters are compared. CO2 and O2 are retrieved using a single prior profile with the PROFFIT code. It is shown that these gases can be retrieved with reasonable SNR using both beamsplitters and that only a small bias exists between these two measurement series.

Major comments: 1. As clear from the abstract and introduction, a major motivation for this work is to enable the Mexico UNAO group to develop XCO2 capability and join TCCON. I thus suggest that the group expand the scope of this investigation to include

processing of their spectra (at least the ones obtained with the CaF2 beamsplitter) using the standard TCCON processing code. Both Hase and Blumenstock run TCCON sites and are thus fully versed in the mechanics of assisting in this extension of scope. Given the clear desire of Baylon et al to join the network, such a modest expansion of scope will thus serve additionally to provide additional knowledge transfer from the KIT group to the Mexico group.

2. Likely not unrelated to 1), the observed diurnal dependence of XCO2 (Fig 10) is almost certainly a result of airmass dependent bias in the retrievals. TCCON processing includes an attempt to account for such bias. Thus, I expect that the TCCON retrievals will substantially reduce the airmass dependence shown in Fig. 10 and additionally alter the seasonal structures (modestly).

Minor comment:

1. Ln 17. Changes in H2O vapor is likely close to changes in CO2 in net change in radiative feedback over past decades.

2. Ln 99. When the KBr UNAO spectra are processed with TCCON software, suggest using same continuum model described by Kiel et al.

3. Ln 185-187 Please explain more fully how the bias (DeltaCO2 = -0.030+- 0.070%) is translated into the scaling factor (0.9986).

4. Ln 200. This is (at best) a hypothesis. Given the (relatively) low biomass in the area, I'm exceedingly doubtful. (see above major comment 2).

---

## Author Comment (AC1) · 15 Mar 2017

We are very grateful for the comments received from both reviewers and each point is addressed below individually:

**Answers to comments from Referee #1**

**1) Is there some reference that supports the validity of comparing the total column amount of CO2, XCO2, with the in situ CO2 amount , as shown in Figure 6. I believe that when averaging kernels and error covariances are taken into account there is little valid overlap. See Rodgers and Connor, JGR, 108, 4116,**

**2003.**

The $CO_2$ in situ time series in MLO is used here to characterize a global background atmosphere at 19°N latitude. The behavior of the surface concentration can be assumed to be valid as the lifetime of $CO_2$ is sufficient long. The following sentence was modified in lines 168-172 in order to clarify that a direct comparison between MLO (in situ) and Altzomoni (FTIR) was not sought in this exercise: "Although both data sets show a similar behavior, the intention of using MLO in this context was to investigate the relative bias between both beamsplitter ensembles with a common reference by arranging the data into bins."

The sentence in lines 203-205 was also changed to: "For comparison, the detrended diurnal cycle of the MLO data is also shown in the figure and despite that it is at a different location and corresponds to in situ measurements, it shows a similar behavior with a smaller amplitude and a minimum which occurs 3 hours later."

**2) Since one of the main objectives of the work is to compare the error contributions of KBr and CaF2 beamsplitters, which of the error sources in Table 2 might have some physical justification for being different between the two beamsplitters. In Table 3 the largest differences in error for CO2 between the splitters seems to be in the noise and solar line contributions. Is there a reason for this?**

Temperature and solar lines are expected to have small contributions to the errors and these differences depend mostly on the chosen ensembles, but the only error source that is expected to differ significantly because of the beamsplitter is the noise. This is due to the fact that the SNR changes with the optical set-up as it can be inferred from Figure 1. Errors due to the solar lines in the case of $CO_2$ are less than 0.01%, the differences in percentage in both the Statistical and the Systematic contributions
might seem big but they are rather small compared to the other sources. The following sentence was added in line 125: "The largest error difference between beamsplitters is expected to originate from the noise for a given ensemble, since the spectral windows used for the retrievals are measured with different signal/noise ratios (see Fig. 1)."

**Minor points**

**Line 7 define XCO2**

Done

**Line 17 water vapor contributes more than any other gas to radiative forcing**

The sentence in lines 16-17 was changed to: "... contributing more than any other anthropogenic gas to the positive total radiative forcing of the Earth ..."

**Line 24 it might be good to add some numbers to describe where the near infrared is, various readers may have various ideas about what is near or far (also line 90)**

The sentence in lines 24-25 was changed to: "... in the near infrared (NIR, 3,300-13,000 cm$^{-1}$) spectral region ..."

We also changed line 90: "... spectra in the mid-infrared (MIR, 200-3,300 cm$^{-1}$) region ..."

**Line 55 /characterisitc/characteristic/**

Done

**Line 59 /begun/began/**

Done

**Line 62 /is part/has been part/**

Done

**Line 104 comment on how effective the removal of the baseline curvature is wrt the retrieval**

The removal of the baseline curvature with the radiometric calibration has very low impact on the retrieval, the average change in the total columns of $CO_2$ and $O_2$ for three days of measurements (85 measurements) is +0.021 ± 0.00086 % and +0.0053 ± 0.00033 %, respectively. This might be due to the 20 baseline points used in the retrieval code, similar to what was recommended by Kiel et al., 2016.

Lines 101-104 were changed to: "The baseline curvature in the spectral region around the dip introduced by the KBr beamsplitter is removed using a simplified radiometric calibration, assuming that the tungsten lamp produces a blackbody spectrum (T=1700 K) and that there is no self-emission of the optical set-up in the spectral region above 4000 $cm^{-1}$. This calibration has a low impact on the columns (+0.021% for $CO_2$, +0.0053% for $O_2$) due to the simultaneous fit of the baseline in the retrieval code."

**Line 115 /to assess/an assessment of/**

Done

**Line 117 /determine where does/determining where/**

Done

**Line 120 /allows to/allows one to/**

Done

**Line 154 /ensamble/ensemble/ also in Line 168,215**

Done

**Line 246 2 also Line 257, 287**

Done

**Line 254 n/a-n/a also Line 301, 310**

Done

**Figure 5. The reference that we found mentioning the TCCON precision goal (0.25) is Wunch et al., 2011.**

The caption in Fig. 5 was changed to: "The black lines depict the existing precision goals for $CO_2$ found in the literature."

**Answers to comments from Referee #2 (P. O. Wennberg)**

**Major comments:**

**1. As clear from the abstract and introduction, a major motivation for this work is to enable the Mexico UNAO group to develop XCO2 capability and join TCCON. I thus suggest that the group expand the scope of this investigation to include processing of their spectra (at least the ones obtained with the CaF2 beamsplitter) using the standard TCCON processing code. Both Hase and Blumenstock**

**run TCCON sites and are thus fully versed in the mechanics of assisting in this extension of scope. Given the clear desire of Baylon et al to join the network, such a modest expansion of scope will thus serve additionally to provide additional knowledge transfer from the KIT group to the Mexico group.**

We agree with the reviewer that a site like Altzomoni, with the available instrumentation, has the potential to join the TCCON network and contribute to global carbon cycle studies. We will look into the possibilities to develop the capability in Mexico to use the community code GFIT in the near future. However, the intention of this work is to present an assessment of the precisions of CO2, $O_2$ and $X_{CO_2}$ obtained when using both beamsplitters so that we can prove the value of our measurements and use this data with confidence for reporting and analyzing $X_{CO_2}$ variability in the region. This is relevant to the community since several NDACC instruments use a KBr beamsplitter and could easily add-on their capability by including an InGaAs detector in their measurement routines. Moreover, doing routine measurements with the $CaF_2$ beamsplitter and analyzing the data according to the TCCON guidelines would require extra resources in funding and manpower which at the moment we have in limited amounts.

**2. Likely not unrelated to 1), the observed diurnal dependence of XCO2 (Fig 10) is almost certainly a result of airmass dependent bias in the retrievals. TCCON processing includes an attempt to account for such bias. Thus, I expect that the TCCON retrievals will substantially reduce the airmass dependence shown in Fig. 10 and additionally alter the seasonal structures (modestly).**

This is a very valid point. At the moment we cannot provide precise evidence that an airmass dependence is not contributing to the diurnal variability which we observe (presented as an average curve in Figure 10). As shown in Dohe, 2013 and Kiel et al., 2016 when using PROFFIT as the processing code, a refined treatment of the background continuum level and, for the case of $O_2$ retrievals, using a detailed model

of the $O_2$ CIA band can reduce the airmass dependence of $CO_2$ and $O_2$ retrievals. We followed this recommendations for the analysis of our data set so we believe that an airmass dependent bias might not be responsible for all the diurnal variability of $X_{CO_2}$ observed at Altzomoni.

The TCCON strategy fits a set of time and zenith angle dependent functions which would assume a symmetrical dependence, a correction which would not fully remove the observed daily dependence at Altzomoni (see Fig.1). This tropical mountain site is surrounded by forests covering large areas and can also be influenced by polluted air which reaches the site in the afternoon (Ochoa et al., 2012), favoring the argument that carbon capture and transport processes are dominating the diurnal variability. This is also supported by what is observed from collocated in situ $CO_2$ measurements.

Thus, following the reviewer's comment, we have removed the sentence in line 200 which stated that $CO_2$ was lowered in average 1.5 ppm only through photosynthesis. The following sentence was added in its place: "Although a treatment of the airmass dependence for $X_{CO_2}$ has been considered following Dohe, 2013 and Kiel et al., 2016, this may still not be fully corrected in the reported $X_{CO_2}$. However, a qualitative analysis correlating these observations with in situ measurements indicate that carbon capture and transport processes are responsible for most of the diurnal variability observed over the Altzomoni site."

Both topics focusing on the reprocessing with the community code GFIT and the development of an airmass correction at Altzomoni will be addressed in future studies.

**Minor comment:**

**1. Ln 17. Changes in H2O vapor is likely close to changes in CO2 in net change in radiative feedback over past decades.**

The sentence in lines 16-17 was changed to: "... contributing more than any other anthropogenic gas to the positive total radiative forcing of the Earth ..."

**2. Ln 99. When the KBr UNAO spectra are processed with TCCON software, suggest using same continuum model described by Kiel et al.**

Yes, we followed the recommendation of Kiel et al., 2016. See answers above.

**3. Ln 185-187 Please explain more fully how the bias (DeltaCO2 = -0.030+-0.070%) is translated into the scaling factor (0.9986).**

The scaling factor for $XCO_2$ (0.9986) comes from the fact that the bias obtained was of $+0.14 \pm 0.064$ % for KBr-CaF$_2$ differences.

**4. Ln 200. This is (at best) a hypothesis. Given the (relatively) low biomass in the area, I'm exceedingly doubtful. (see above major comment 2).**

An answer to this point is given above.

**References**

C. Ochoa, D. Baumgardner, M. Grutter, J. Allan, J. Fast, and B. Rappenglueck. Physical and chemical properties of the regional mixed layer of Mexico's megapolis part ii: evaluation of measured and modeled trace gases and particle size distributions. Atmospheric Chemistry and Physics, 12(21):10161–10179, 2012.

Sussane Dohe. Measurements of atmospheric $CO_2$ columns using ground-based FTIR spectra. PhD thesis, Karlsruher Institut für Technologie (KIT), 2013.

M. Kiel, D. Wunch, P. O. Wennberg, G. C. Toon, F. Hase, and T. Blumenstock. Improved retrieval of gas abundances from near-infrared solar FTIR spectra measured at the Karlsruhe TCCON station. Atmospheric Measurement Techniques, 9(2):669–682, 2016.

---

## Author Response (AR2)

Dear Hal Maring,

We have carefully read the additional comments made by Reviewer 2, and have prepared the following answer and the changes are reflected in the attached revised manuscript.

**Answers to comments from Referee #2**

1. Abstract (line 8). 0.56 ppm in XCO2 is not a small bias. In fact this bias alone would consume the entire error budget of TC-CON. In fact, site-to-site bias of 0.56 ppm would essentially render TCCON useless for most carbon cycle science. The precision requirements described (Rayner and OBrien; Olsen and Randerson) refer to bias-free measurement precision and so are not relevant to the discussion here where a one hour average of the FTS data will have essentially negligibly small random precision. Given the uncertainty in the cause of the bias between the two beamsplitters, it isnt clear how it could be homoginized without extensive and ongoing characterization. In summary, I think this study rather than confirming that KBr beamsplitter is acceptable offers significant evidence that this option should be approached cautiously.

We agree with this comment and removed the word "small" in line 8 of the abstract. Part of the next sentence was also left out "...and could be taken into account when homogenizing or comparing data from both beamsplitters".

The sentence "For the homogenization ..." that starts in line 188 was also removed.

We realize that this bias in not insignificant and should be treated cautiously.

2. I continue to object to the comparison between the diurnal changes in XCO2 shown here and in situ measurements from MLO. This makes absolutely no sense and the implication that this comparison gives confidence in the XCO2 diurnal signal is ill advised. What is the flux implied by the changes in XCO2? Answer: huge! I urge the authors to scrub this from the manuscript. Fine to show the diurnal figure, but I believe that the changes shown are (largely) just evidence for airmass bias. A plot of XCO2 vs airmass (which should certainly be included) would illustrate that the majority of the diurnal signal is associated with airmass rather than time of day (as would be expected from a flux).

We followed the recommendation and removed Figure 10 and replaced it with another one showing the dependence of our measurements with the SZA. The text was adapted as followed:

Removed the sentence in lines 199 to 201 and replaced it with the following text:

"In Figure 10, we present the XCO2 and seasonally detrended XCO2 averages showing a clear dependence with respect to the solar zenith angle."

The following argument was added to denote a large CO2 diurnal variability observed in in situ data and its implications to the SZA dependence:

"However, a quantitative analysis correlating these observations with in situ measurements at Altzomoni, with a night-to-day average amplitude of approximately 5 ppm, indicates that carbon capture processes can be contributing significantly to the shown SZA dependence".

Finally, the sentence in lines 208 to 211 was removed "For comparison, ..."

**3. What is the basis for the final sentence: This data set confirms a small influence of anthropogenic emissions especially during the afternoon, when the regional boundary layer reaches the height of the station.? A figure of XCO might offer some evidence for this, but that is not discussed or shown.**

The second part of the sentence and next sentence in lines 237 to 239 was completely removed: "..., while the mean diurnal pattern reflects a decrease of up to 1.5 ppm during sunlit hours. This data set confirms a small influence of anthropogenic emissions especially during the afternoon, when the regional boundary layer reaches the height of the station".

**Errata**

"Intercomparision" in line 50 was corrected to "intercomparison"

"weren't" in line 168 was corrected to "were not"

The sentence that begins in line 184, that reads "The bias for CO2 column was calculated from the biases obtained above ( $\Delta XCO2$  and  $O2 \Delta O2$ ) using ..." should read "The bias for CO2 column was calculated from the biases obtained above ( $\Delta XCO2$  and  $\Delta O2$ ) using ...".

"Remaines" in line 219 was corrected to "remains"

"beamsplktters" in line 223 was corrected to "beamsplitters"

The first sentence of the Figure 5 caption ends with "... obtained from the top-down approach." but should end with "... obtained from the Section 3.2 approach." In the second sentence of the same caption the word "precisin" was corrected to "precision".

[revised manuscript text omitted]